# Levels and determinants of quality antenatal care in Bangladesh: Evidence from the Bangladesh Demographic and Health Survey

Ema Akter[1]*, Aniqa Tasnim Hossain[1], Ahmed Ehsanur Rahman[1], Anisuddin Ahmed[1], Tazeen Tahsina[1], Tania Sultana Tanwi[1], Nowrin Nusrat[1], Quamrun Nahar[1], Shams El Arifeen[1], Mahbub Elahi Chowdhury[2]

1 Maternal and Child Health Division (MCHD), International Centre for Diarrhoeal Disease Research, Bangladesh (icddr,b), Dhaka, Bangladesh, 2 Health System and Population Studies Division (HSPSD), International Centre for Diarrhoeal Disease Research, Bangladesh (icddr,b), Dhaka, Bangladesh

* eakter@isrt.ac.bd, ema.akter@icddrb.org

## Abstract

**Data Availability Statement:** The data set can be found from the Demographic and Health Surveys

### Background

Assessing the quality of antenatal care (ANC) is imperative for improving care provisions during pregnancy to ensure the health of mother and baby. In Bangladesh, there is a dearth of research on ANC quality using nationally representative data to understand its levels and determinants. Thus, the current study aimed to assess ANC quality and identify the sociode-mographic factors associated with the usage of quality ANC services in Bangladesh.

### Methods

Secondary data analysis was conducted using the last two Bangladesh Demographic and Health Surveys (BDHSs) from 2014 and 2017–18. A total of 8,277 ever-married women were included in the analysis (3,631 from 2014 and 4,646 from 2017–18). The quality ANC index was constructed using a principal component analysis on the following ANC components: weight and blood pressure measurements, blood and urine test results, counselling about pregnancy complications and completion of a minimum of four ANC visits, one of which was performed by a medically trained provider. Multinomial logistic regression was used to determine the strength of the association.

### Results

The percentage of mothers who received all components of quality ANC increased from about 13% in 2014 to 18% in 2017–18 ($p < 0.001$). Women from the poorest group, those in rural areas, with no education, a high birth order and no media exposure were less likely to receive high-quality ANC than those from the richest group, those from urban areas, with a higher level of education, a low birth order and media exposure, respectively.

(DHS) program website URL: https://dhsprogram.
com/data/available-datasets.cfm.

**Funding:** The author(s) received no specific
funding for this work.

**Competing interests:** The authors have declared
that no competing interests exist.

## Conclusion

Although the quality of ANC improved from 2014 to 2017–18, it remains poor in Bangladesh.
Therefore, there is a need to develop targeted interventions for different socio-demographic
groups to improve the overall quality of ANC. Future interventions should address both the
demand and supply-side perspectives.

## Introduction

Antenatal care (ANC) refers to the care given to women during their pregnancy. Assessing
ANC quality is imperative for improving care provisions during pregnancy to ensure the
health of mother and baby [1]. Systematic supervision of the mother during pregnancy is the
core intervention within the continuum of such care [2]. Complications during pregnancy are
considered a leading cause of maternal death, stillbirth and neonatal death, for which poor-
quality ANC is a contributing factor [3]. The latest Sustainable Development Goals include the
target of reducing global maternal deaths to less than 70 per 100,000 live births by 2030 [4]. It
has been established that appropriate-quality ANC can save lives [3] and lower maternal mor-
tality by up to 20% [5, 6]. Quality ANC has a protective effect against adverse pregnancy out-
comes, such as low birth weight and premature birth [7–9]. In addition, utilisation of skilled
birth attendants (i.e. doctors, nurses and midwives) during delivery and postnatal care
increases with quality ANC [10]. Although 62% of pregnant women worldwide attended at
least four WHO-recommended ANC activities in 2017 [11], the corresponding figure for Ban-
gladesh was only 47% (as per the Bangladesh Demographic and Health Survey [BDHS]) in
2017–18 [12].

Assessment of ANC quality requires information on service usage of the recommended
contents of the care [13–17]. The literature has defined quality ANC in terms of its required
components, which should be delivered during pregnancy for a better, healthier life [3, 12, 18,
19]. Components that constitute quality ANC can differ among countries [20]. The 2016
WHO ANC model recommends several interventions to upgrade the delivery of quality ANC,
including nutritional interventions, maternal and foetal assessments, preventive measures,
physiological symptom interventions and health system interventions [3]. Surveys have been
conducted in Bangladesh to ask women about services received during ANC, such as weight
and blood pressure measurements, blood and urine tests, ultrasonogram tests, medical advice
regarding signs of pregnancy complications and postpartum family planning. These criteria,
along with the criterion of a minimum of four visits completed during ANC, have been used
to define the concept of quality ANC [12, 21–23]. However, only 18% receive quality ANC in
Bangladesh, as documented by the BDHS 2017–18 [12]. Determining ANC quality enables
medical practitioners to identify risk factors and deliver appropriate treatment [24]. The qual-
ity of care given to pregnant women in areas with few resources is a significant concern.

According to a study conducted in Bangladesh, the technical content of ANC received by
mothers during their care visits is suboptimal in most facilities, particularly those in rural set-
tings [25]. Additionally, women are usually unable to complete four or more ANC visits dur-
ing their pregnancy due to a lack of access to healthcare facilities and skilled health providers
[26]. An estimated 99% of Bangladeshi healthcare facilities offer ANC services, but only 4% are
sufficiently prepared to provide quality ANC services in accordance with the WHO guidelines,
as per the Bangladesh Health Facility Survey 2017 [27]. Appropriate provision of ANC during
pregnancy is evidenced by reduced maternal mortalities and afflictions as well as healthier

motherhood experiences with better maternal health outcomes [28]. Therefore, all pregnant women should have access to maternal health services, and it is beneficial to examine the factors related to these services to minimise pregnancy-related mortality and morbidity [29].

A recently published study by Singh et al. (2019) reveals that little research has been conducted to understand the factors that influence ANC quality in low- and middle-income countries [20]. There have been few studies on ANC in Bangladesh, most of which have been conducted in rural contexts [26, 30–34]. The BDHS, which is nationally representative survey, is conducted once every three years and collects information on ANC. Only, a few researchers have used existing demographic and health survey data to examine the determinants and contents of ANC in Bangladesh [6, 25, 34–37]. A study conducted by Rahman et al. (2017) explored the determinants of four or more ANC utilisation using the BDHS 2011 and 2014, where they found the level of education, place of residence and wealth status as the determinants [25]. Other studies using the BDHS 2014 to examine the factors associated with ANC have revealed that educated women and those with the highest wealth status are more likely to receive ANC [6, 35–37]. A study conducted using the BDHS 2007 explored the scenario of ANC services in a rural context and reports that mothers' education and mass media exposure have a significant effect on their ANC utilisation [34].

To the best of the authors' knowledge, there is a dearth of research assessing the determinants of quality ANC receival. Such studies are important to guide interventions in specific areas, not only to increase ANC coverage but also to increase the quality ANC coverage. Thus, this study aims to extend the evidence in this area by determining ANC quality using the definition mentioned in the latest BDHS 2017–18 using the last two BDHS surveys (2014 and 2017–18) and examining the factors associated with the usage of quality ANC.

## Materials and methods

Secondary data analysis was conducted using data from the last two BDHSs (2014 and 2017–18), which were the seventh and eighth in the series. They covered all districts across the administrative divisions of Bangladesh through nationally representative cross-sectional surveys [12, 21]. As a sampling frame, the surveys used a list of enumeration areas (EAs) from the 2011 Population and Housing Census of the People's Republic of Bangladesh provided by the Bangladesh Bureau of Statistics.

The BDHS is a retrospective study following a multistage stratified cluster sampling design. In the first stage, EAs are selected based on probability proportional to EA size. Then, a complete household listing is conducted in all selected EAs to provide a sampling frame for the second-stage household selection. In the second stage, a systematic sample of households per EA is selected to obtain statistically reliable estimates of key demographic and health variables. Next, ANC-related information is collected for the last live birth for women who had given birth within the three years preceding the survey.

Considering the survey weight, a total of 8,277 ever-married women with their most recent live birth were included in the analysis for BDHSs 2014 and 2017–18, which included 3,631 and 4,646 women, respectively (Fig 1). Information was gathered about each individual; the number of ANC visits they completed and the components delivered in the ANC.

The quality ANC index was constructed using the 2014 and 2017–18 BDHSs, following the definition of quality ANC given in BDHS 2017–18 considering the following components: weight and blood pressure measurements, blood and urine tests, counselling about pregnancy complications and number of ANC visits completed. For the ANC to be considered good quality, four visits must have taken place, one of which must have been performed by a medically trained provider. The quality ANC index was created via a principal component analysis

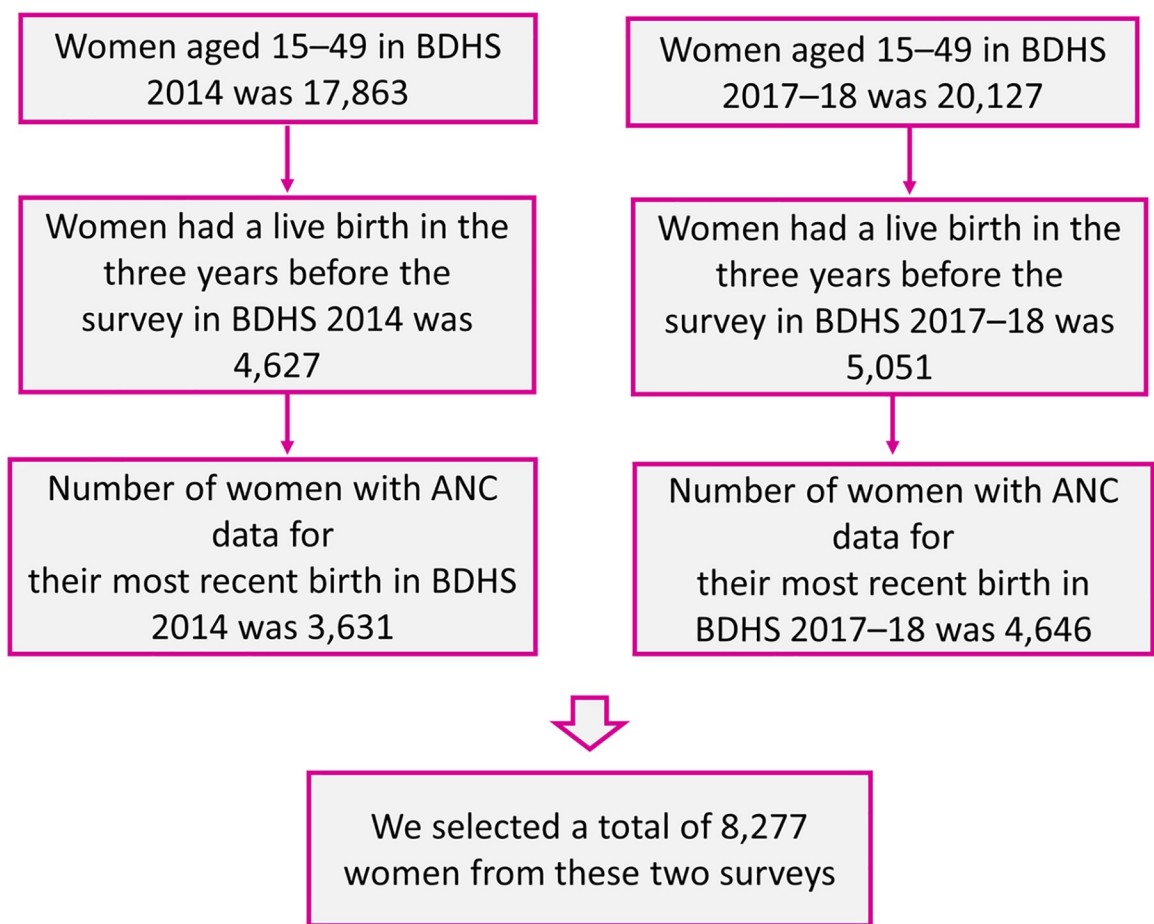

**Fig 1. Flowchart of the sample selection process.**

(PCA); principal components are new variables constructed from linear combinations of the original variables. Here, the first principal component was considered the index obtained from the PCA. It explained most of the variances in the data. This component was divided into three equal parts. It was a categorical variable with three categories (low, middle and high) and was considered as the outcome variable used in this study, i.e. the quality of ANC tertile.

Several socioeconomic and demographic characteristics were considered as explanatory variables. These included division (Dhaka, Barishal, Chattogram, Khulna, Rajshahi, Rangpur, Sylhet and Mymensingh), place of residence (urban or rural), wealth quintile (richest, richer, middle, poorer and poorest; derived using PCA based on ownership of household goods), education level (higher, primary, secondary and no education), maternal age in years (15–19, 20–24, 25–29, 30–34 and 35–49), birth order (1, 2–3, 4–5 and 6+), religion (Islam or others) and media exposure (exposed or unexposed). Respondents were considered exposed to media if they performed any of the following three actions at least once a week: reading a newspaper or magazine, watching television or listening to the radio. There were seven administrative divisions (Dhaka, Barishal, Chattogram, Khulna, Rajshahi, Rangpur and Sylhet) in survey year 2014, whereas there were eight divisions (Dhaka, Barishal, Chattogram, Khulna, Rajshahi, Rangpur, Sylhet and Mymensingh) in survey year 2017–18. Mymensingh was reported with Dhaka for survey year 2014.

## Statistical analysis

Descriptive statistics were used to report the characteristics of the mothers accessing quality ANC. Moreover, the *p*-value from a proportion test was reported for both surveys to observe the difference in the ordered characteristics of quality ANC according to socioeconomic and demographic factors. A 5% level of significance was considered to indicate a significance difference. The *p*-value from a proportion test used to observe the trend in quality ANC received over the surveyed periods was also reported. Multinomial logistic regression was performed for the pooled data from the surveys to assess the factors affecting the usage of quality ANC, considering the low- quality category of ANC as a reference. A crude and adjusted odds ratio (OR) and a corresponding 95% confidence interval (CI) were reported for this regression. The results were reported considering the surveys' complex design and sample weights. The statistical package Stata 14 [38] was used to perform the analysis.

## Results

As seen in Table 1, the Dhaka division had the highest percentage of women (31%) participating in the survey, while the Barishal and Mymensingh divisions had the lowest percentage (5%). Around 71% of the mothers were from rural areas, and 17% were from the poorest socioeconomic group. In both surveys, more than half of the mothers (51%) had a secondary level of education. The majority of the mothers (35%) were in the 20- to 24-year age group. A large number (48%) reported a birth order of 2–3. Considering religion, around 92% of the mothers were Muslim. Around 58% of the mothers were exposed to media.

The percentage of women receiving all six components of quality ANC increased from about 13% (95% CI: 12%–15%) in 2014 to 18% (95% CI: 16%–19%) in 2017–18 ($p < 0.001$; Fig 2). Among the ANC components, blood pressure measurement was most commonly received by the mothers, with about 88% receiving it in 2014 and 93% receiving it in 2017–18. The second most commonly received component was weight measurement at around 84% in 2014 and 88% in 2017–18. There was also an increase in the provision of the rest of the ANC components (except for counselling services about pregnancy complications) from 2014 to 2017–18.

As illustrated in Table 2, the results indicated a significant difference of quality ANC receival by type of residence, wealth status, education level, birth order and media exposure. The percentage of mothers in rural areas receiving high-quality ANC increased by approximately 15% between the two BDHSs (17% in 2014 and 32% in 2017–18). While a significant proportion of mothers with no education received low-quality ANC (64% in 2014 and 57% in 2017–18). From 2014 to 2017–18, the percentage of mothers receiving high-quality ANC increased for all age groups and birth orders. Moreover, the percentage of those exposed to media who received high-quality ANC increased by around 15% between the surveys (31% in 2014 and 46% in 2017–18).

Table 3 displays the results of the multivariate analysis of the socioeconomic and demographic factors associated with the mothers who received quality ANC. The mothers who answered the 2017–18 BDHS had a higher likelihood of receiving high-quality ANC compared with those who answered the 2014 survey. Over time, the number of mothers receiving quality ANC increased significantly. The results showed that place of residence, wealth status, educational level, birth order and media exposure were significant determinants for the quality of ANC index.

Mothers living in rural areas were less likely to receive high-quality ANC compared with those in urban areas (Adjusted Odds Ratio [AOR]: 0.78; 95% CI: 0.67–0.90). Furthermore, the mothers from the poorest group were less likely to receive high-quality ANC compared with those from the richest group (AOR: 0.13; 95% CI: 0.10–0.17). Mothers with no

**Table 1.  Distribution of the mothers according to background characteristics, presented as frequency and percentage.**

| Background characteristics | Survey year 2014 | Survey year 2017–18 | Total |
|---|---|---|---|
| | n (%) | n (%) | n (%) |
| **Division** | | | |
| Dhaka | 1,371 (37.8) | 1,204 (25.9) | 2,575 (31.1) |
| Barisal | 196 (5.4) | 245 (5.3) | 441 (5.3) |
| Chattogram | 754 (20.8) | 974 (21.0) | 1,728 (20.9) |
| Khulna | 327 (9.0) | 446 (9.6) | 773 (9.3) |
| Rajshahi | 350 (9.7) | 556 (12.0) | 906 (11.0) |
| Rangpur | 365 (10.1) | 507 (10.9) | 872 (10.5) |
| Sylhet | 267 (7.4) | 327 (7.1) | 595 (7.2) |
| Mymensingh | - | 388 (8.3) | 388 (4.7) |
| **Place of residence** | | | |
| Urban | 1,081 (29.8) | 2,366 (28.6) | 2,366 (28.6) |
| Rural | 2,549 (70.2) | 3,362 (72.4) | 5,911 (71.4) |
| **Wealth status** | | | |
| Wealthiest | 869 (23.9) | 975 (21.0) | 1,844 (22.3) |
| Wealthy | 856 (23.6) | 977 (21.0) | 1,833 (22.1) |
| Middle class | 717 (19.8) | 908 (19.5) | 1,625 (19.6) |
| Poor | 614 (16.9) | 929 (20.0) | 1,543 (18.6) |
| Poorest | 574 (15.8) | 858 (18.5) | 1,432 (17.3) |
| **Educational level** | | | |
| Higher | 450 (12.4) | 854 (18.4) | 1,304 (15.8) |
| Secondary | 1,910 (52.6) | 2,340 (50.4) | 4,250 (51.4) |
| Primary | 898 (24.7) | 1,219 (26.2) | 2,117 (25.6) |
| No education | 373 (10.3) | 234 (5.0) | 607 (7.3) |
| **Age in years** | | | |
| 15–19 | 784 (21.6) | 856 (18.4) | 1,639 (19.8) |
| 20–24 | 1,225 (33.8) | 1,638 (35.3) | 2,863 (34.6) |
| 25–29 | 927 (25.5) | 1,204 (25.9) | 2,131 (25.7) |
| 30–34 | 486 (13.4) | 693 (14.9) | 1,179 (14.3) |
| 35–49 | 208 (5.7) | 256 (5.5) | 465 (5.6) |
| **Birth order** | | | |
| 1 | 1,563 (43.1) | 1,855 (39.9) | 3,418 (41.3) |
| 2–3 | 1,683 (46.3) | 2,290 (49.3) | 3,972 (48.0) |
| 4–5 | 317 (8.7) | 432 (9.3) | 749 (9.1) |
| 6+ | 69 (1.9) | 70 (1.5) | 138 (1.7) |
| **Religion** | | | |
| Islam | 3,354 (92.4) | 4,258 (91.6) | 7,612 (92.0) |
| Others | 277 (7.6) | 388 (8.4) | 665 (8.0) |
| **Exposure to media** | | | |
| Exposed | 2,121 (58.4) | 2,680 (57.7) | 4,801 (58.0) |
| Unexposed | 1,510 (41.6) | 1,967 (42.3) | 3,476 (42.0) |

education were less likely to receive high-quality ANC compared with those with a higher level of education (AOR: 0.20; 95% CI: 0.15–0.28). Compared with older mothers, younger mothers had a lower likelihood of receiving high-quality ANC. Mothers with a birth order of 6+ had a lower likelihood of receiving high-quality ANC compared with those with a

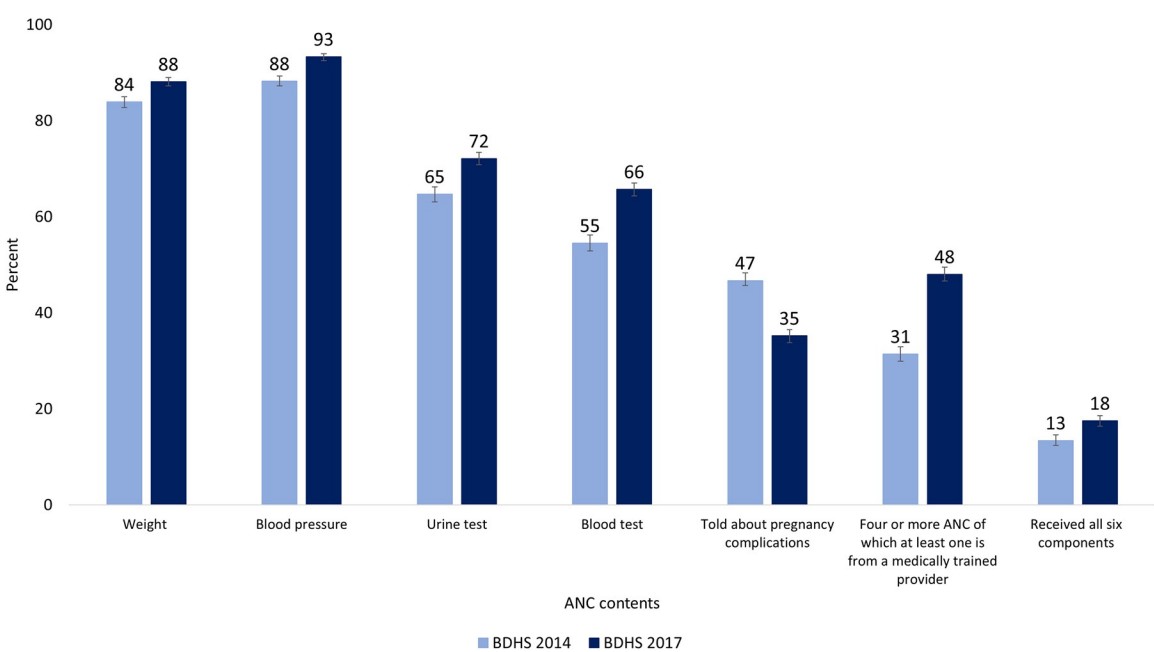

**Fig 2. Percentage of mothers receiving specific antenatal services.**

birth order of 1 (AOR: 0.45; 95% CI: 0.25–0.82). Mothers who were not exposed to media were less likely to receive high-quality ANC compared with those exposed to media (AOR: 0.65; 95% CI: 0.56–0.75).

Fig 3 presents the odds ratio values with 95% CI for the interaction between the place of residence and wealth quintile. The mothers who lived in urban areas and belonged to the poorest group had a lower probability of receiving high-quality ANC than those who lived in urban areas and belonged to the richest group (OR: 0.06; 95% CI: 0.03–0.09). Moreover, the mothers who lived in rural areas and were from the poorest group had significantly lower odds of receiving high-quality ANC than those who lived in urban areas and were from the richest group (OR: 0.05; 95% CI: 0.04–0.06).

## Discussion

This study revealed that socioeconomic and demographic characteristics are related to receiving better-quality ANC and that a considerable proportion of women do not receive adequate-quality ANC. A low percentage of the mothers surveyed in the two BDHSs reviewed received all six components of quality ANC, with a significant difference observed between the survey results (13% in 2014 BDHS and 18% in 2017–18 BDHS). Women with a high birth order, lower levels of education, poorer economic status, rural living status and lack of media exposure were less likely to receive high-quality ANC than those with a low birth order, higher levels of education, higher economic status, urban living status and media exposure, respectively. Approximately 57% of the women from the richest group and 57% of those with a higher education level received high-quality ANC, whereas only around 12% of those from the poorest group and 12% of those with no education did.

This study's finding that the mothers with a higher level of education had a higher likelihood of receiving quality ANC was consistent with other studies' findings from different parts of the world [6, 37, 39–42]. One study conducted in Bangladesh documented that mothers'

**Table 2. Descriptive statistics regarding quality of ANC received by background characteristic, presented as frequency and percentage.**

| Background characteristics | Quality of antenatal care index | | | | | | | |
|---|---|---|---|---|---|---|---|---|
| | Survey year 2014 | | | | Survey year 2017–18 | | | |
| | Low | Medium | High | P-value | Low | Medium | High | P-value |
| | n (%) | n (%) | n (%) | | n (%) | n (%) | n (%) | |
| **Division** | | | | | | | | |
| Dhaka | 605 (44.1) | 410 (29.9) | 356 (26.0) | <0.001 | 353 (29.3) | 304 (25.3) | 547 (45.4) | <0.001 |
| Barisal | 86 (44.0) | 73 (37.4) | 36 (18.6) | <0.001 | 74 (30.2) | 86 (35.3) | 84 (34.5) | 0.337 |
| Chattogram | 296 (39.2) | 292 (38.7) | 167 (22.1) | <0.001 | 325 (33.3) | 322 (33.0) | 328 (33.6) | 0.885 |
| Khulna | 148 (45.2) | 103 (31.4) | 76 (23.3) | <0.001 | 125 (28.1) | 127 (28.6) | 193 (43.3) | <0.001 |
| Rajshahi | 185 (52.8) | 104 (29.8) | 61 (17.5) | <0.001 | 224 (40.2) | 145 (26.0) | 188 (33.8) | 0.022 |
| Rangpur | 183 (50.1) | 116 (31.9) | 65 (18.0) | <0.001 | 200 (39.6) | 132 (26.0) | 174 (34.4) | 0.083 |
| Sylhet | 112 (41.8) | 97 (36.4) | 58 (21.8) | <0.001 | 113 (34.6) | 116 (35.5) | 98 (29.9) | 0.213 |
| Mymensingh | - | - | - | - | 182 (47.1) | 90 (23.3) | 115 (29.6) | <0.001 |
| **Place of residence** | | | | | | | | |
| Urban | 358 (33.1) | 329 (30.4) | 395 (36.5) | 0.092 | 299 (23.3) | 322 (25.1) | 663 (51.6) | <0.001 |
| Rural | 1,257 (49.3) | 868 (34.0) | 425 (16.7) | <0.001 | 1,298 (38.6) | 1,000 (29.8) | 1,064 (31.6) | <0.001 |
| **Wealth quintile** | | | | | | | | |
| Richest | 180 (20.7) | 280 (32.3) | 409 (47.0) | <0.001 | 113 (11.6) | 214 (22.0) | 648 (66.4) | <0.001 |
| Richer | 321 (37.5) | 312 (36.5) | 223 (26.0) | <0.001 | 253 (25.9) | 306 (31.3) | 418 (42.8) | <0.001 |
| Middle | 385 (53.7) | 229 (32.0) | 103 (14.3) | <0.001 | 349 (38.4) | 263 (29.0) | 296 (32.6) | 0.008 |
| Poorer | 342 (55.6) | 215 (35.0) | 58 (9.4) | <0.001 | 434 (46.8) | 277 (29.8) | 218 (23.4) | <0.001 |
| Poorest | 387 (67.3) | 160 (27.8) | 28 (4.9) | <0.001 | 447 (52.1) | 263 (30.6) | 148 (17.3) | <0.001 |
| **Education level** | | | | | | | | |
| Higher | 89 (19.9) | 139 (31.0) | 221 (49.2) | <0.001 | 128 (15.0) | 200 (23.4) | 526 (61.6) | <0.001 |
| Secondary | 792 (41.5) | 664 (34.7) | 454 (23.8) | <0.001 | 762 (32.6) | 662 (28.3) | 916 (39.2) | <0.001 |
| Primary | 496 (55.2) | 298 (33.2) | 104 (11.6) | <0.001 | 574 (47.1) | 394 (32.4) | 250 (20.6) | <0.001 |
| No education | 237 (63.6) | 95 (25.5) | 41 (10.9) | <0.001 | 133 (56.8) | 67 (28.5) | 34 (14.7) | <0.001 |
| **Age in years** | | | | | | | | |
| 15–19 | 397 (50.7) | 253 (32.4) | 133 (17.0) | <0.001 | 298 (34.9) | 259 (30.3) | 298 (34.9) | 1.000 |
| 20–24 | 519 (42.4) | 395 (32.3) | 311 (25.4) | <0.001 | 540 (33.0) | 485 (29.6) | 614 (37.5) | 0.006 |
| 25–29 | 402 (43.4) | 305 (33.0) | 219 (23.6) | <0.001 | 397 (33.0) | 332 (27.6) | 475 (39.5) | 0.001 |
| 30–34 | 203 (41.8) | 187 (38.4) | 96 (19.8) | <0.001 | 261 (37.7) | 174 (25.1) | 258 (37.2) | 0.864 |
| 35–49 | 92 (44.2) | 55 (26.5) | 61 (29.4) | 0.001 | 100 (39.0) | 74 (28.8) | 83 (32.3) | 0.111 |
| **Birth order** | | | | | | | | |
| 1 | 638 (40.8) | 509 (32.6) | 416 (26.6) | <0.001 | 525 (28.3) | 532 (28.7) | 797 (43.0) | <0.001 |
| 2–3 | 751 (44.6) | 574 (34.1) | 358 (21.3) | <0.001 | 812 (35.4) | 651 (28.4) | 827 (36.1) | 0.638 |
| 4–5 | 185 (58.5) | 91 (28.8) | 40 (12.7) | <0.001 | 222 (51.5) | 120 (27.8) | 90 (20.8) | <0.001 |
| 6+ | 41 (59.4) | 23 (32.8) | 5 (7.8) | <0.001 | 38 (53.7) | 19 (27.8) | 13 (18.5) | <0.001 |
| **Religion** | | | | | | | | |
| Islam | 1,514 (45.1) | 1,098 (32.7) | 742 (22.1) | <0.001 | 1,493 (35.1) | 1,202 (28.2) | 1,564 (36.7) | 0.103 |
| Others | 101 (36.3) | 99 (35.6) | 78 (28.1) | 0.038 | 104 (26.8) | 121 (31.1) | 164 (42.1) | <0.001 |
| **Exposed to media** | | | | | | | | |
| Exposed | 751 (35.4) | 706 (33.3) | 664 (31.3) | 0.005 | 698 (26.0) | 738 (27.5) | 1,244 (46.4) | <0.001 |
| Unexposed | 864 (57.2) | 490 (32.5) | 156 (10.3) | <0.001 | 899 (45.7) | 585 (29.7) | 483 (24.6) | <0.001 |

education level strongly affects optimal uptake of ANC services [6]. Educated women are more conscious about healthcare for themselves and their newborns from the beginning of pregnancy, and as a result, they are willing to access such care [37]. In addition, they have greater decision-making power concerning socioeconomic matters in their households, which gives

**Table 3. Factors associated with quality of antenatal care received.**

| Background characteristics | Middle-quality antenatal care | | | High-quality antenatal care | | |
|---|---|---|---|---|---|---|
| | n (%) | Crude odds ratio (95% confidence interval) | Adjusted odds ratio (95% confidence interval) | n (%) | Crude odds ratio (95% confidence interval) | Adjusted odds ratio (95% confidence interval) |
| **Survey year** | | | | | | |
| 2014 | 1,196 (33.0) | 1.00 | 1.00 | 820 (22.6) | 1.00 | 1.00 |
| 2017–18 | 1,323 (28.5) | 1.12 (1.01–1.24) | 1.23 (1.10–1.37) | 1,727 (37.2) | 2.13 (1.91–2.37) | 2.70 (2.38–3.06) |
| **Division** | | | | | | |
| Dhaka | 715 (27.8) | 1.00 | 1.00 | 903 (35.1) | 1.00 | 1.00 |
| Barisal | 160 (36.2) | 1.34 (1.05–1.70) | 1.63 (1.27–2.08) | 121 (27.4) | 0.80 (0.62–1.03) | 1.37 (1.03–1.83) |
| Chattogram | 613 (35.5) | 1.33 (1.14–1.54) | 1.31 (1.12–1.53) | 494 (28.6) | 0.85 (0.73–0.98) | 0.87 (0.73–1.03) |
| Khulna | 230 (30.0) | 1.13 (0.92–1.38) | 1.17 (0.95–1.45) | 269 (34.9) | 1.05 (0.86–1.27) | 1.25 (1.01–1.56) |
| Rajshahi | 249 (27.5) | 0.82 (0.68–0.98) | 0.87 (0.72–1.05) | 249 (27.5) | 0.65 (0.54–0.78) | 0.79 (0.64–0.98) |
| Rangpur | 248 (28.5) | 0.87 (0.72–1.05) | 1.04 (0.85–1.27) | 240 (27.5) | 0.66 (0.55–0.80) | 1.14 (0.91–1.42) |
| Sylhet | 214 (35.9) | 1.27 (1.03–1.57) | 1.53 (1.23–1.91) | 156 (26.3) | 0.74 (0.59–0.92) | 1.25 (0.96–1.62) |
| Mymensingh | 90 (23.3) | 0.66 (0.51–0.87) | 0.73 (0.55–0.97) | 115 (29.6) | 0.67 (0.52–0.86) | 0.80 (0.60–1.07) |
| **Place of residence** | | | | | | |
| Urban | 651 (27.5) | 1.00 | 1.00 | 1,059 (44.7) | 1.00 | 1.00 |
| Rural | 1,868 (31.6) | 0.74 (0.65–0.84) | 1.05 (0.91–1.21) | 1,489 (25.2) | 0.36 (0.32–0.41) | 0.78 (0.67–0.90) |
| **Wealth quintile** | | | | | | |
| Richest | 495 (26.8) | 1.00 | 1.00 | 1056 (57.3) | 1.00 | 1.00 |
| Richer | 618 (33.7) | 0.64 (0.53–0.77) | 0.70 (0.57–0.84) | 640 (34.9) | 0.31 (0.26–0.37) | 0.41 (0.34–0.49) |
| Middle | 492 (30.3) | 0.40 (0.33–0.48) | 0.44 (0.36–0.54) | 399 (24.5) | 0.15 (0.13–0.18) | 0.22 (0.18–0.27) |
| Poorer | 492 (31.9) | 0.38 (0.31–0.45) | 0.46 (0.37–0.57) | 276 (17.9) | 0.10 (0.08–0.12) | 0.17 (0.13–0.21) |
| Poorest | 422 (29.5) | 0.30 (0.25–0.36) | 0.40 (0.31–0.50) | 176 (12.3) | 0.06 (0.05–0.07) | 0.13 (0.10–0.17) |
| **Education level** | | | | | | |
| Higher | 339 (26.0) | 1.00 | 1.00 | 747 (57.3) | 1.00 | 1.00 |
| Secondary | 1,325 (31.2) | 0.55 (0.45–0.66) | 0.72 (0.60–0.88) | 1,370 (32.2) | 0.26 (0.22–0.30) | 0.52 (0.43–0.62) |
| Primary | 693 (32.7) | 0.42 (0.34–0.50) | 0.66 (0.53–0.81) | 355 (16.8) | 0.10 (0.08–0.12) | 0.27 (0.21–0.33) |
| No education | 162 (26.7) | 0.28 (0.22–0.36) | 0.48 (0.36–0.64) | 75 (12.4) | 0.06 (0.04–0.08) | 0.20 (0.15–0.28) |
| **Age in years** | | | | | | |
| 15–19 | 513 (31.3) | 0.95 (0.79–1.14) | 0.71 (0.56–90) | 431 (26.3) | 0.81 (0.68–0.98) | 0.60 (0.46–0.77) |

*(Continued)*

**Table 3.** (Continued)

| Background characteristics | Middle-quality antenatal care | | | High-quality antenatal care | | |
|---|---|---|---|---|---|---|
| | n (%) | Crude odds ratio (95% confidence interval) | Adjusted odds ratio (95% confidence interval) | n (%) | Crude odds ratio (95% confidence interval) | Adjusted odds ratio (95% confidence interval) |
| 20–24 | 880 (30.7) | 1.07 (0.91–1.26) | 0.77 (0.56–0.90) | 924 (32.3) | 1.15 (0.97–1.35) | 0.70 (0.56–0.87) |
| 25–29 | 637 (29.9) | 1.03 (0.86–1.22) | 0.87 (0.72–1.05) | 694 (32.6) | 1.14 (0.96–1.35) | 0.92 (0.75–1.13) |
| 30–34 | 360 (30.6) | 1.00 | 1.00 | 354 (30.0) | 1.00 | 1.00 |
| 35–49 | 129 (27.7) | 0.86 (0.67–1.12) | 1.00 (0.76–1.33) | 144 (31.0) | 0.98 (0.76–1.27) | 1.41 (1.03–1.92) |
| **Birth order** | | | | | | |
| 1 | 1,041 (30.5) | 1.00 | 1.00 | 1,213 (35.5) | 1.00 | 1.00 |
| 2–3 | 1,225 (30.8) | 0.88 (0.78–0.98) | 0.84 (0.72–0.97) | 1,186 (29.9) | 0.73 (0.65–0.81) | 0.69 (0.59–0.81) |
| 4–5 | 211 (28.2) | 0.58 (0.48–0.70) | 0.55 (0.43–0.71) | 130 (17.3) | 0.31 (0.25–0.38) | 0.36 (0.27–0.48) |
| 6+ | 42 (30.3) | 0.60 (0.41–0.88) | 0.67 (0.43–1.06) | 18 (13.2) | 0.22 (0.13–0.38) | 0.45 (0.25–0.82) |
| **Religion** | | | | | | |
| Islam | 2,300 (30.2) | 1.00 | 1.00 | 2,306 (30.3) | 1.00 | 1.00 |
| Others | 219 (33.0) | 1.40 (1.15–1.71) | 1.34 (1.09–1.65) | 241 (36.3) | 1.54 (1.27–1.87) | 1.39 (1.11–1.73) |
| **Exposed to media** | | | | | | |
| Exposed | 1,444 (30.1) | 1.00 | 1.00 | 1,909 (39.8) | 1.00 | 1.00 |
| Unexposed | 1,075 (30.9) | 0.61 (0.55–0.68) | 0.84 (0.74–0.95) | 639 (18.4) | 0.27 (0.25–0.31) | 0.65 (0.56–0.75) |

them the confidence and ability to make decisions about their health and the use of quality care services [40]. Furthermore, educated women have better knowledge of the importance of ANC during pregnancy.

Differences in the receival of quality ANC were also observed in relation to place of residence. Women from urban areas had a higher likelihood of receiving high-quality ANC, whereas those from rural areas had a lower likelihood. A study conducted in Nepal had similar findings, documenting that mothers from rural areas are less interested in accessing ANC services than mothers in urban regions [39]. This may be attributed to the fact that unskilled healthcare providers are providing services in rural areas, meaning women there have a lower likelihood of receiving good-quality ANC [41]. The discrepancy may also be the result of lack of accessibility due to transportation or other supply-side restrictions (e.g. a lack of medical facilities, qualified medical staff or accurate diagnostic laboratory test findings) [37]. In addition, the lower socioeconomic status of women in rural areas might be associated with lower uptake of ANC services [6].

A prime determining factor of quality ANC receival is wealth status. This study found that women from the poorest group were less likely to receive high-quality ANC compared with those from the richest group, consistent with the findings of previous research [6, 20, 41]. A study conducted using 91 national household surveys documented that poor women receive lower quality of care [1]. Costly healthcare services may be a reason; women of a lower wealth status receive poor-quality ANC. Without overall betterment of their standard of living, efforts

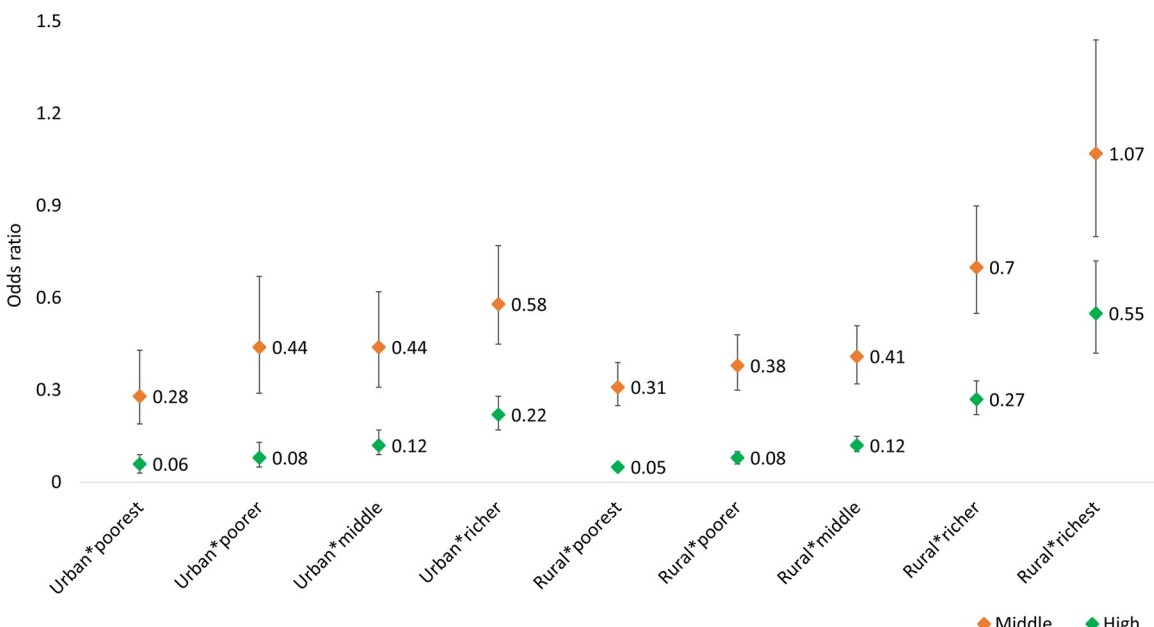

**Fig 3. Odds ratio for the interaction between place of residence and wealth quintile (Urban\*richest as reference category).**

to encourage poorer mothers' utilisation of maternal health services might not attain useful outcomes [6].

Age also affects women's uptake of ANC services. This study found that the young mothers had a lower likelihood of accessing high-quality ANC. However, older women in Bangladesh frequently access ANC services [35]; they are more likely to undergo difficulties at birth and have higher uptake of such care [28]. They may also have more awareness and knowledge relating to ANC. In a 2021 study, Tessema and Minyihun [43] documented that older women understand the benefits of visiting healthcare facilities.

This study also found that birth order is related to quality ANC uptake. The mothers with higher birth orders were less likely to utilise high-quality ANC services, implying that women with previous birth experience demand less ANC. This finding is similar to the results obtained from the 2013 study conducted by Nketiah et al. [28] documenting that in the case of later births, women are less likely to access ANC. However, in earlier pregnancies, women may experience various difficulties, which contributes to their need for such services. In Bangladesh, it is apparent that women giving birth for the first time have a higher likelihood of using recommended ANC [37].

This study's findings on media exposure as a factor influencing uptake of ANC services were also in line with those of previous research [6, 37, 44, 45]. For instance, a study conducted in Uganda reported that women who have access to media are more likely to use ANC services than women who do not [45]. They are more informed than their counterparts about the usefulness of ANC services during pregnancy and ask ANC providers for care that may help them lead a healthy life [45]. Additionally, mothers who watch television are more likely to be aware of the problems related to pregnancy and the importance of ANC [6].

## Strengths and limitations

We believe that this is the first study designing an ANC quality tertile using components of ANC services utilised by women in Bangladesh from nationally representative demographic

and health surveys, meaning that the findings are generalisable at the national and sub-national levels. In addition, this study was conducted using the pooled data of the 2014 and 2017–18 BDHSs indicates a larger population-based study for examining the factors associated with quality ANC uptake.

The data were collected for three years preceding the survey using the recall method; as a result, there may be some recall bias in the study. However, in the sample, 36.8% of the births were identified to have occurred within the previous year, 34.8% within the past 1–2 years and 28.4% within the past 2–3 years [46]. As the recall period varied, the recall bias should have been low. In the surveys, ANC-related information was only available for mothers with live births. Therefore, stillbirths and mothers who died during pregnancy or delivery were not included in the study. In addition, data were not available for all of the WHO 2016 ANC model recommendations. However, the BDHS provides the best available population-based nationally representative data on ANC quality in Bangladesh. Due to data limitations, several important confounding factors that may influence uptake of quality ANC could not be addressed, such as cost of care, availability of healthcare facilities and knowledge about maternal healthcare services.

## Conclusion

Although there was an improvement in the number of Bangladeshi mothers receiving quality ANC from 2014 to 2017–18, this number remains low. Women with a low education level, rural living status, low socioeconomic status, a high birth order and lack of media exposure have been identified as receiving low-quality ANC. This demands improvement of ANC quality via the development of targeted interventions for these groups. Future interventions must consider both the demand and supply-side perspectives. An education programme for women with regular knowledge-enhancing sessions for pregnant mothers may play a vital role in increasing the awareness of the importance of completing ANC visits. Moreover, documentaries about maternal and child healthcare could be regularly broadcast on television, YouTube, Facebook and radio stations. Simultaneously, the supply side should be strengthened by reaching out to the target groups regarding the provision of health services with the application of mobile health technology along with information and communication technology. Trained healthcare providers at the field level with sufficient logistical support should be effectively engaged to provide quality ANC to pregnant mothers via the organisation of satellite and mobile clinics as appropriate based on the local context. If the findings and suggestions presented in this paper are acted upon, there will be improvement in ANC quality in Bangladesh.

## Supporting information

**S1 File.**
(DOCX)

## Acknowledgments

Many thanks to the National Institute of Population Research and Training (NIPORT) and the Monitoring and Evaluation to Assess and Use Results Demographic and Health Surveys (MEASURE DHS) for giving their permission to utilise the BDHS dataset.

## Author Contributions

**Conceptualization:** Ema Akter, Ahmed Ehsanur Rahman, Mahbub Elahi Chowdhury.

**Formal analysis:** Ema Akter, Aniqa Tasnim Hossain, Anisuddin Ahmed, Nowrin Nusrat.

**Methodology:** Ema Akter, Ahmed Ehsanur Rahman, Tazeen Tahsina, Tania Sultana Tanwi, Quamrun Nahar, Shams El Arifeen, Mahbub Elahi Chowdhury.

**Software:** Ema Akter.

**Supervision:** Mahbub Elahi Chowdhury.

**Writing – original draft:** Ema Akter.

**Writing – review & editing:** Ema Akter, Aniqa Tasnim Hossain, Ahmed Ehsanur Rahman, Anisuddin Ahmed, Tazeen Tahsina, Tania Sultana Tanwi, Nowrin Nusrat, Quamrun Nahar, Shams El Arifeen, Mahbub Elahi Chowdhury.

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
