## [Decision Letter · Decision Letter 0]

19 Sep 2022

PONE-D-22-15496Levels and determinants of quality of antenatal care in Bangladesh: Evidence from the Bangladesh Demographic and Health SurveyPLOS ONE

Dear Dr. Akter,

Thank you for submitting your manuscript to PLOS ONE. After careful consideration, we feel that it has merit but does not fully meet PLOS ONE’s publication criteria as it currently stands. Therefore, we invite you to submit a revised version of the manuscript that addresses the points raised during the review process.

Please note that we have only been able to secure a single reviewer to assess your manuscript. We are issuing a decision on your manuscript at this point to prevent further delays in the evaluation of your manuscript. Please be aware that the editor who handles your revised manuscript might find it necessary to invite additional reviewers to assess this work once the revised manuscript is submitted. However, we will aim to proceed on the basis of this single review if possible. 

The reviewer’s comments may be seen below. While the reviewer has recommended discussion around the novelty of the study, PLOS ONE does not issue decisions based on perceived novelty, and do not require revisions on this aspect of the reviewer’s comments. However, the reviewer has noted there are several similar studies within this topic area, we would therefore require related literature to be discussed within the body of the mansucript, and clarification of how results from the current study further contributes to scientific understanding. Our second publication criterion notes that "If a submitted study replicates or is very similar to previous work, authors must provide a sound scientific rationale for the submitted work and clearly reference and discuss the existing literature. Submissions that replicate or are derivative of existing work will likely be rejected if authors do not provide adequate justification." (https://journals.plos.org/plosone/s/criteria-for-publication#loc-2).

Could you please revise the manuscript to carefully address the concerns raised?

We look forward to receiving your revised manuscript.

Kind regards,

Lucinda Shen, MSc

Staff Editor

PLOS ONE

Journal Requirements:

"We are grateful to National Institute of Population Research and Training (NIPORT), Bangladesh, and Monitoring and Evaluation to Assess and Use Results Demographic and Health Surveys (MEASURE DHS), USA for allowing permission to utilise the BDHS dataset."

Reviewers' comments:

Reviewer's Responses to Questions

**Comments to the Author**

1. Is the manuscript technically sound, and do the data support the conclusions?

Reviewer #1: Partly

2. Has the statistical analysis been performed appropriately and rigorously? 

Reviewer #1: Yes

3. Have the authors made all data underlying the findings in their manuscript fully available?

Reviewer #1: Yes

4. Is the manuscript presented in an intelligible fashion and written in standard English?

Reviewer #1: Yes

5. Review Comments to the Author

Reviewer #1: Thanks to the authors for addressing an important issue in the context of Bangladesh although several studies already have been done using the same dataset. But I found it interesting and innovative in terms of study objectives.

I must be thanking to the authors.

However I have some addressable comments regarding the paper as a reviewer perspective.

Abstract: fine no comments from me. Please rewrite the conclusion here it has now a general discussion please be specific based on your findings.

Background: please standardised the rationality and novelty of the paper and set the rationality corresponding to other similar studies.

Methods: well organized.

Results l: clear, understandable.

Discussion: overtly written please be specific when set the logical presentation of the findings relying on relevant literature.

Conclusion: please sperate recommendations from conclusion. Although it is done based on cross sectional data so recommendations are not necessary.

6. PLOS authors have the option to publish the peer review history of their article (what does this mean?). If published, this will include your full peer review and any attached files.

Reviewer #1: **Yes: **Md. Ashfikur Rahman

---

## [Author Response · Author response to Decision Letter 0]

31 Oct 2022

Reviewers' comments:

Comments to the Author 

Abstract: fine no comments from me. Please rewrite the conclusion here it has now a general discussion please be specific based on your findings.

Response: Thank you for your suggestion. We have updated it as “Although there has been an improvement in the quality of ANC from 2014 to 2017/18, it is still of poor standard in Bangladesh. There is a need of developing targeted interventions for different socio-demographic groups to improve the overall quality of ANC. Future interventions should address both the demand and supply side perspectives.”

Background: please standardised the rationality and novelty of the paper and set the rationality corresponding to other similar studies.

Response: Thanks. We have reviewed the studies related to antenatal care (ANC). There are several studies regarding ANC, of which most of them have been conducted in rural context. Bangladesh Demographic and Health Survey (BDHS) has been conducted every three years which is a nationally representative survey. Also, to our knowledge there is a dearth of study which explored status of quality ANC recently mentioned in BDHS 2017/18. After your suggestion, we have added several statements mentioning how this study contributing to the new knowledge.

“There are several studies on ANC in Bangladesh, of which most of them have been conducted in rural context (24, 29-33). The Demographic and Health Survey has been conducted once every three years, which is a nationally representative survey. Only a few researchers used existing demographic and health survey data to examine the determinants and contents of ANC contacts in Bangladesh (7, 23, 33-36). A study by Rahman et al. (2017) using BDHS 2011 and 2014 explored the determinants of four or more ANC utilisation where the level of education, place of residence and wealth status were found as the determinants (23). Studies using BDHS (2014) to examine the factors associated with ANC, revealed that educated women and women from the richest wealth status were more likely in having ANC (7, 34-36). Another study conducted using BDHS 2007 explored the scenario of ANC services in a rural context and reported that education of the mothers, and being exposed to mass media have a significant effect on having ANC (33). To the best of our knowledge, there is a dearth of study assessing the determinants in receiving the quality of ANC.”

Methods: well organized.

Response: Thank you.

Results l: clear, understandable.

Response: Thank you for your comment.

Discussion: overtly written please be specific when set the logical presentation of the findings relying on relevant literature.

Response: Thanks. We have updated according to your comments in the discussion part. 

Conclusion: please sperate recommendations from conclusion. Although it is done based on cross sectional data so recommendations are not necessary.

Response: Thank you. We have updated it.

---

## [Decision Letter · Decision Letter 1]

10 Jan 2023

PONE-D-22-15496R1Levels and determinants of quality of antenatal care in Bangladesh: Evidence from the Bangladesh Demographic and Health SurveyPLOS ONE

Dear Dr. Akter,

Thank you for submitting your manuscript to PLOS ONE. After careful consideration, we feel that it has merit but does not fully meet PLOS ONE’s publication criteria as it currently stands. Therefore, we invite you to submit a revised version of the manuscript that addresses the points raised during the review process.

The reviewer added several comments. Please address them line by line.  Please ensure that your decision is justified on PLOS ONE’s publication criteria and not, for example, on novelty or perceived impact.

We look forward to receiving your revised manuscript.

Kind regards,

Md Nuruzzaman Khan

Academic Editor

PLOS ONE

Reviewers' comments:

Reviewer's Responses to Questions

**Comments to the Author**

1. If the authors have adequately addressed your comments raised in a previous round of review and you feel that this manuscript is now acceptable for publication, you may indicate that here to bypass the “Comments to the Author” section, enter your conflict of interest statement in the “Confidential to Editor” section, and submit your "Accept" recommendation.

Reviewer #2: All comments have been addressed

2. Is the manuscript technically sound, and do the data support the conclusions?

Reviewer #2: Partly

3. Has the statistical analysis been performed appropriately and rigorously? 

Reviewer #2: Yes

4. Have the authors made all data underlying the findings in their manuscript fully available?

Reviewer #2: Yes

5. Is the manuscript presented in an intelligible fashion and written in standard English?

Reviewer #2: No

6. Review Comments to the Author

Reviewer #2: Track changed file

Introduction

Overall: It would be better to start introduction with a sentence about ANC. It would be better to improve the language extensively that would help the readers to understand it clearly.

Line 53-56: Please rewrite the sentence to increase the readability. Another statement, “…. less than 70 per 100,000 live births”

Line 57: “176 per 100,000 live births”, I am not sure which year statistics it is.

Line 62-64: I think it’s better to show the comparative statistics of same period otherwise it could mislead the information. The time period 2007-14, where we were targeted to achieve MDG but from 2017-18 till now on, we are targeted to achieve SDG and our strategies have also been changed.

Line 68-81: I would strongly suggest the authors to rewrite this section in a way describing the status and pathways of ANC quality and how it is contributing to improving the maternal health situation.

Line 82-88: I would say it’s not a part of introduction, it could be a better fit in the methodology instead.

I think the rationale still needs an improvement with proper articulation of study importance, existing evidence, and gaps.

Methods

Overall: I can see the authors mentioned about the number of visits, but it is not clear how it was categorized in the analysis. Language editing is recommended. In some places, it’s not clear enough what the author wanted to explain.

Line 119: I think it’s an analysis of secondary data, so I would rather prefer to write it as “analysis of secondary data”.

Line 122: I would suggest the author to describe how the clustering was done and the sample was drawn, in brief. I think it’s available in the BDHS reports.

Line 124-125: The authors have mentioned about sampling weight. I would prefer to have a brief in the analysis section how the weighting was considered. Was it just women individual weight (v005) or they have considered the complex survey design consider the psu and strata as well or they have considered all women factor (in all forms, residence cluster, education cluster, etc.), although all women factors will not an applicable here.

Line 137 – 139 is not clear enough to me. The authors have done PCA analysis to construct ANC index, that’s a good way to construct this kind of index, but I would prefer to have a clearer insight of the process in brief as it will help the non-statistical reader to understand or get a sense of the way of analysis. They would not feel like lost in somewhere middle of nowhere. It’s also needed to include the categorization process.

Line 153-156: It would be better to rewrite the analysis section more precisely. The authors have mentioned about trend, I did not find any trend analysis in the result section. What they have done is that they have analysed two survey datasets separately. Readers would rather be keen to know about the change over the years.

Just a simple discussion or point to be noted that, for a higher prevalence of an event its better not to fit logistic regression as it gives better stable result when the event prevalence is lower like around 10%.

Results

Table 1: The interpretation of the results is clear, but I would like to suggest two things. It would be interesting to show a trend analysis.

Table 2: In terms of divisional derivation, I think in 2017 data, ANC quality in Dhaka and Mymensingh have some misrepresentation in way like in 2014 data this two division were reported combined. This is why we can see Dhaka division has a major improvement in terms of seeking quality ANC services compared to other divisions. I would also suggest the authors to check some of the findings presented here, I mean p-values. To me, it seems some to the findings have a fluctuation to get a significant result. I could also be wrong, but please just check.

Table 3: I am not sure how the survey year is a factor that would be associated with quality of antenatal care received? And how the authors have run this analysis? And how the authors have adjusted the whole model with it? Division variable is again misleading the findings here. Interpretation would be written in more specific and clear way. I would prefer to write Birth order instead of Birth rate. Birth rate means different issue, definition “Birth rate is the number of individuals born in a population in a given amount of time.”

Overall: Authors have interpreted some insignificant findings instead of significant results. Need major revision.

Discussion

Line 232-233: I didn’t find the significance testing for the difference between two surveys in terms of mothers received all six components of quality ANC.

Line 230-238: Please rewrite this section representing your major findings.

Overall: It would be better if the author improves the discussion part further, discuss the findings rather repeating the results again. In some section, the discussion needs to be more precise and relevant with the findings. It required substantial revisions and English editing as well to make the findings more understandable.

7. PLOS authors have the option to publish the peer review history of their article (what does this mean?). If published, this will include your full peer review and any attached files.

Reviewer #2: No

---

## [Author Response · Author response to Decision Letter 1]

31 Mar 2023

To

The Editor,

PLOS ONE

Dear Editor,

Thank you very much for your time and allowing us to submit our revised manuscript titled “Levels and determinants of quality antenatal care in Bangladesh: Evidence from the Bangladesh Demographic and Health Survey” in your reputed journal. We have addressed all the issues kindly raised by the respected reviewer. We are really thankful to the respected reviewer for their valuable comments that immensely helped to improve the quality of the manuscript.

Sincerely yours,

Ema Akter

Maternal and Child Health Division, icddr,b (International Centre for Diarrhoeal Disease Research, Bangladesh), Dhaka, Bangladesh 

//////////////////////////////////////////////////////////////////////////////////////////////////////////////

Reviewers' comments:

Comments to the Author 

Introduction

Overall: It would be better to start introduction with a sentence about ANC. It would be better to improve the language extensively that would help the readers to understand it clearly.

Response: Thank you reviewer for your suggestion. We have updated the Introduction as per your suggestion.

Line 53-56: Please rewrite the sentence to increase the readability. Another statement, “…. less than 70 per 100,000 live births”

Response: Thanks. We have modified it as “Complications during pregnancy are considered a leading cause of maternal death, stillbirth and neonatal death, for which poor-quality ANC is a contributing factor (3). The latest Sustainable Development Goals include the target of reducing global maternal deaths to less than 70 per 100,000 live births by 2030 (4).”

Line 57: “176 per 100,000 live births”, I am not sure which year statistics it is.

Response: Thanks so much for this comment. Sorry for our mistakes. We have submitted it earlier, however, now, we are in a new year. We have updated this in the manuscript.

Line 62-64: I think it’s better to show the comparative statistics of same period otherwise it could mislead the information. The time period 2007-14, where we were targeted to achieve MDG but from 2017-18 till now on, we are targeted to achieve SDG and our strategies have also been changed.

Response: Yes, we have updated it as “Although 62% of pregnant women worldwide attended at least four WHO-recommended ANC activities in 2017 (11), the corresponding figure for Bangladesh was only 47% (as per the Bangladesh Demographic and Health Survey [BDHS]) in 2017–18 (12).”

Line 68-81: I would strongly suggest the authors to rewrite this section in a way describing the status and pathways of ANC quality and how it is contributing to improving the maternal health situation.

Response: Thanks. We have rewritten according to your comments.

Line 82-88: I would say it’s not a part of introduction, it could be a better fit in the methodology instead.

Response: Thank you. We have updated it.

I think the rationale still needs an improvement with proper articulation of study importance, existing evidence, and gaps.

Response: Thanks. We have worked on this part.

Methods

Overall: I can see the authors mentioned about the number of visits, but it is not clear how it was categorized in the analysis. Language editing is recommended. In some places, it’s not clear enough what the author wanted to explain.

Response: Number of visits of antenatal care (ANC) was used in the definition of quality ANC. The latest BDHS (2017–18) defined quality ANC as follows: woman has four or more ANC visits, of which at least one is with a medically trained provider, and receives all of the basic components of ANC (weight and blood pressure measurements, urine and blood tests, and information on signs of possible complications) at least once. Using these above-mentioned components, we calculated quality ANC index.

Thanks. We have done editing of the statements.

Line 119: I think it’s an analysis of secondary data, so I would rather prefer to write it as “analysis of secondary data”.

Response: Thanks for your comment. We have updated it as “Secondary data analysis was conducted using data from the last two BDHSs (2014 and 2017–18), which were the seventh and eighth in the series.”

Line 122: I would suggest the author to describe how the clustering was done and the sample was drawn, in brief. I think it’s available in the BDHS reports.

Response: Yes, it’s available in BDHS reports. We have updated it in our manuscript.

Line 124-125: The authors have mentioned about sampling weight. I would prefer to have a brief in the analysis section how the weighting was considered. Was it just women individual weight (v005) or they have considered the complex survey design consider the psu and strata as well or they have considered all women factor (in all forms, residence cluster, education cluster, etc.), although all women factors will not an applicable here.

Response: Thanks for your comment. In our manuscript, results were reported considering complex survey design and sample weights of the surveys. We have written it in the analysis section.

Line 137 – 139 is not clear enough to me. The authors have done PCA analysis to construct ANC index, that’s a good way to construct this kind of index, but I would prefer to have a clearer insight of the process in brief as it will help the non-statistical reader to understand or get a sense of the way of analysis. They would not feel like lost in somewhere middle of nowhere. It’s also needed to include the categorization process.

Response: Many thanks for your suggestion. We have updated this part. 

Line 153-156: It would be better to rewrite the analysis section more precisely. The authors have mentioned about trend, I did not find any trend analysis in the result section. What they have done is that they have analysed two survey datasets separately. Readers would rather be keen to know about the change over the years.

Response: Thanks. Here, our main outcome variable is quality ANC. We observed the trend in receiving ANC quality over the last two surveys. In the result section, we have mentioned “The percentage of women receiving all six components of quality ANC increased from about 13% (95% CI: 12%–15%) in 2014 to 18% (95% CI: 16%–19%) in 2017–18 (p < 0.001; Fig. 2)” in the result section to show the change over the years.

Just a simple discussion or point to be noted that, for a higher prevalence of an event its better not to fit logistic regression as it gives better stable result when the event prevalence is lower like around 10%.

Response: Here, our outcome variable is quality of ANC index which has three categories (low, middle and high). Each category has around equal number of observation as we calculated this index using principal component analysis where we divided the component into three equal parts.

Results

Table 1: The interpretation of the results is clear, but I would like to suggest two things. It would be interesting to show a trend analysis.

Response: Thank you. We have done this. As our interested variable is quality ANC, we observed the trend in receiving ANC contacts and components over the last two surveys (Figure 2). The latest BDHS 2017-18 defined the quality ANC and BDHS 2014 reported the same components as BDHS 2017-18 which we used to make the quality ANC index. We mentioned in our manuscript that “The percentage of women receiving all six components of quality ANC increased from about 13% (95% CI: 12%–15%) in 2014 to 18% (95% CI: 16%–19%) in 2017–18 (p < 0.001; Fig. 2)” to report the change over the years in receiving quality ANC.

Table 2: In terms of divisional derivation, I think in 2017 data, ANC quality in Dhaka and Mymensingh have some misrepresentation in way like in 2014 data this two division were reported combined. This is why we can see Dhaka division has a major improvement in terms of seeking quality ANC services compared to other divisions. I would also suggest the authors to check some of the findings presented here, I mean p-values. To me, it seems some to the findings have a fluctuation to get a significant result. I could also be wrong, but please just check.

Response: Many thanks. We have mentioned the divisional variation in the method section now. However, there is signification difference in receiving quality ANC in Dhaka for both the surveys (BDHS 2014 and BDHS 2017-18). 

We checked the p-values and got the reported results. This p-values comes from proportion test for observing the difference over ordered characteristics of quality ANC, according to socioeconomic and demographic characteristics. 

Table 3: I am not sure how the survey year is a factor that would be associated with quality of antenatal care received? And how the authors have run this analysis? And how the authors have adjusted the whole model with it? Division variable is again misleading the findings here. Interpretation would be written in more specific and clear way. I would prefer to write Birth order instead of Birth rate. Birth rate means different issue, definition “Birth rate is the number of individuals born in a population in a given amount of time.”

Response: Thanks. Here we appended the two surveys together. We labeled these two datasets as 2014 and 2017-18 in the pooled data. We consider the survey year as a factor to observe the difference in receiving the quality ANC over the time if we keep the all other covariates as constant. We found that there happened significant increase in receiving the quality ANC over the time. We have rewritten the interpretation of the result section.

Thank you so much. We have updated the name as “Birth order” instead of “Birth rate”.

Overall: Authors have interpreted some insignificant findings instead of significant results. Need major revision.

Response: Thanks. We have updated this in our manuscript.

Discussion

Line 232-233: I didn’t find the significance testing for the difference between two surveys in terms of mothers received all six components of quality ANC.

Response: We have mentioned about this testing in the analysis section as well as in the result section (Fig 2). We added this as “The p-value from a proportion test used to observe the differences in quality ANC received over the surveyed periods was also reported” in the analysis section and “The percentage of women receiving all six components of quality ANC increased from about 13% (95% CI: 12%–15%) in 2014 to 18% (95% CI: 16%–19%) in 2017–18 (p < 0.001; Fig. 2).” in the result section. 

Line 230-238: Please rewrite this section representing your major findings.

Response: Thank you. We have updated this part.

Overall: It would be better if the author improves the discussion part further, discuss the findings rather repeating the results again. In some section, the discussion needs to be more precise and relevant with the findings. It required substantial revisions and English editing as well to make the findings more understandable.

Response: Thank you so much. We have updated discussion part according to your comment. In addition to that, the copyediting of this manuscript was conducted by “Proofed Inc”, USA.

---

## [Editor Report · Decision Letter 2]

18 Apr 2023

Levels and determinants of quality antenatal care in Bangladesh: Evidence from the Bangladesh Demographic and Health Survey

PONE-D-22-15496R2

Dear Dr. Akter,

We’re pleased to inform you that your manuscript has been judged scientifically suitable for publication and will be formally accepted for publication once it meets all outstanding technical requirements.

Kind regards,

Md. Nuruzzaman Khan

Academic Editor

PLOS ONE
---

## [Editor Report · Acceptance letter]

25 Apr 2023

PONE-D-22-15496R2 

Levels and determinants of quality antenatal care in Bangladesh: Evidence from the Bangladesh Demographic and Health Survey 

Dear Dr. Akter:

I'm pleased to inform you that your manuscript has been deemed suitable for publication in PLOS ONE. Congratulations! Your manuscript is now with our production department. 

Kind regards, 

on behalf of

Dr. Md. Nuruzzaman Khan 

Academic Editor

PLOS ONE